# Impact of Feed Composition on Rumen Microbial Dynamics and Phenotypic Traits in Beef Cattle

**DOI:** 10.3390/microorganisms13020310

**Published:** 2025-01-31

**Authors:** André L. A. Neves, Ricardo Augusto Mendonça Vieira, Einar Vargas-Bello-Pérez, Yanhong Chen, Tim McAllister, Kim H. Ominski, Limei Lin, Le Luo Guan

**Affiliations:** 1Department of Veterinary and Animal Sciences, Faculty of Health and Medical Sciences, University of Copenhagen, Grønnegårdsvej 3, DK-1870 Frederiksberg C, Denmark; 2Department of Agricultural, Food and Nutritional Science, University of Alberta, Edmonton, AB T6G 2R3, Canada; yanhong@ualberta.ca; 3Laboratório de Zootecnia, Universidade Estadual do Norte Fluminense, Av. Alberto Lamego 2000, Campos dos Goytacazes 28013-602, RJ, Brazil; ramvieira@uenf.br; 4Facultad de Zootecnia y Ecología, Universidad Autónoma de Chihuahua, Chihuahua 31453, Mexico; einar.vargasbelloperez@reading.ac.uk; 5Department of International Development, School of Agriculture, Policy and Development, University of Reading, P.O. Box 237, Earley Gate, Reading RG6 6EU, UK; 6Lethbridge Research Center, Agriculture and Agri-Food Canada, Lethbridge, AB T1J 4B1, Canada; tim.mcallister@agr.gc.ca; 7Department of Animal Science & National Centre for Livestock and the Environment (NCLE), University of Manitoba, Winnipeg, MB R3T 2N2, Canada; kim.ominski@umanitoba.ca; 8Faculty of Land and Food Systems, University of British Columbia, Vancouver, BC V6T 1Z4, Canada; limeil01@mail.ubc.ca

**Keywords:** rumen microbial abundance, feed composition, feed efficiency, methane emissions

## Abstract

The rumen microbiome is central to feed digestion and host performance, making it an important target for improving ruminant productivity and sustainability. This study investigated how feed composition influences rumen microbial abundance and phenotypic traits in beef cattle. Fifty-nine Angus bulls were assigned to forage- and grain-based diets in a randomized block design, evaluating microbial dynamics, methane emissions, and feed efficiency. Quantitative PCR (qPCR) quantified bacterial, archaeal, fungal, and protozoal populations. Grain-based diets reduced bacterial and fungal counts compared to forage diets (1.1 × 10^11^ vs. 2.8 × 10^11^ copies of 16S rRNA genes and 1.5 × 10^3^ vs. 3.5 × 10^4^ copies of 18S rRNA genes/mL, respectively), while protozoan and methanogen populations remained stable. Microbial abundance correlated with feed intake metrics, including dry matter and neutral detergent fiber intakes. Methane emissions were lower in grain-fed bulls (14.8 vs. 18.0 L CH_4_/kg DMI), though feed efficiency metrics showed no direct association with microbial abundance. Comparative analysis revealed adaptive microbial shifts in response to dietary changes, with functional redundancy maintaining rumen stability and supporting host performance. These findings provide insights into how feed composition shapes rumen microbial dynamics and host phenotypes, highlighting the functional adaptability of the rumen microbiome during dietary transitions.

## 1. Introduction

The rumen microbiome plays a central role in feed digestion, animal productivity, and the sustainability of ruminant agriculture. By facilitating the breakdown of fibrous feedstuffs, rumen microbes provide fermentation end-products, including volatile fatty acids and microbial protein, to the host animal [1,2]. However, microbial fermentation in the rumen also produces methane, a potent greenhouse gas that represents both an environmental challenge and an energy loss for the host [3]. Recent advances in microbiome research have highlighted the complexity and dynamic nature of rumen microbial communities, with diet recognized as a primary driver of changes in microbial composition and functional activity [4]. Understanding how dietary inputs influence microbial dynamics is critical for optimizing feed efficiency and mitigating methane emissions in ruminant production systems [5].

Diet composition, particularly the forage-to-concentrate ratio, exerts a profound influence on rumen microbial population dynamics. Forage-based diets are typically rich in fiber and favor the growth of bacteria and fungi specialized in plant cell wall degradation [6]. Conversely, grain-based diets, with their high starch content, often shift microbial communities toward amylolytic populations [7], leading to changes in fermentation pathways and the physico-chemical properties of the rumen environment [8]. Feed composition can alter the balance between hydrogen-producing and hydrogen-utilizing microbes, with downstream effects on methane production and host nutrient utilization [9]. Forage-based diets tend to result in higher methane emissions per unit of DM fermented as compared to grain-based diets, as the fermentation of fibrous feeds generates more hydrogen for methanogenesis [10]. While the impacts of forage-to-grain transitions on rumen microbiota are well-documented [11,12], knowledge gaps remain regarding how their populational shifts interact with host phenotypic traits, such as feed conversion efficiency and methane emissions. Previous research on the assessment of population dynamics during forage-to-grain transitions has predominantly focused on bacteria and has not examined other microbial groups—such as archaea, fungi, and protozoa—individually [13]. Therefore, investigating these relationships can provide insights into the mechanisms underlying host–microbe interactions and their implications for sustainable ruminant production.

It is well established that rumen microbial functions are driven by both prokaryotes (bacteria, archaea) and eukaryotes (protozoa and fungi) [7]. This study aimed to evaluate the impact of feed composition on rumen microbial abundance of four microbial groups and its association with phenotypic traits (e.g., feed intake, feed efficiency, weight gain) and methane emissions in beef cattle. These objectives were addressed by integrating microbial and phenotypic data within a randomized block design. Comparative analysis across feeding periods further highlighted the adaptability of rumen microbial populations to dietary changes.

## 2. Materials and Methods

### 2.1. Animal Study

The experimental procedures described in this study were approved by the Veterinary Services and the Animal Care Committee, University of Manitoba, Canada. Fifty-nine purebred Red and Black Angus bulls (mean age of 249 ± 22 days and average body weight of 314 ± 32 kg) were raised in confinement at the Glenlea Research Station (University of Manitoba) according to the Canadian Council on Animal Care guidelines [14]. The bulls were randomly assigned to four pens, and each pen was bedded with a mixture of barley/flaxseed straw and equipped with GrowSafe^®^ (GrowSafe Systems Ltd., Airdrie, AB, Canada) feed intake system and a heated watering bowl. Bulls were fed forage or grain diets over two experimental periods (Periods 1 and 2, each one with 80-day duration) in a randomized block design (Appendix A). Bulls in pens 1 and 4 were only fed forage or grain diets in both study periods, whereas bulls in pens 2 and 3 were fed forage and grain in Period 1, respectively, and then they were switched to the alternate diet in Period 2 (Appendix A).

Individual feed intake was measured by the GrowSafe^®^ feeding system to provide growth and intake data (DMI) needed for the estimation of feed conversion ratio (FCR), which was used as a measure of feed efficiency in this study. FCR was determined as a ratio of DMI to average daily gain (individual animals) and computed on a biweekly basis [15]. Residual feed intake (RFI) was calculated as the difference between observed feed intake and predicted feed intake, with predictions derived from a regression model based on metabolic body weight (BW^0.75^) and average daily gain (ADG), following the methods described by Koch et al. [16] and further refined by Basarab et al. [17] and Berry and Crowley [18]. The RFI measurement spanned 79 days, during which intake data were rigorously validated to exclude days with feed disappearance below 95%. This approach yielded 76 valid intake days, allowing for a robust assessment of feed efficiency and eliminating potential confounding factors that could arise from inaccurate feed intake measurements. Starch and NDF contents (Appendix A) of the respective diets were used to calculate starch and NDF intakes from the daily DMI [19]. Finally, enteric CH_4_ emissions were measured using the sulfur hexafluoride (SF_6_) tracer gas technique, a method reported by Berndt et al. [20] and Deightonet al. [21]. Methane was measured on four distinct occasions during the adaptation phase to account for variability and allow animals to acclimate to the experimental setup: Days-12, -9, -6, and -2 relative to the start of the RFI period. To avoid interfering with feed intake measurements, CH_4_ collection was deliberately excluded during the RFI phase. Measurements resumed immediately after the RFI period, with three consecutive days of sampling (Days 80, 82, and 85), ensuring post-RFI CH_4_ data collection while minimizing measurement errors.

### 2.2. Rumen Fluid Sampling

Rumen fluid samples were collected using a Geishauser oral probe [22] on Days 0 and 80 in Period 1 and on Days 100 and 180 in Period 2. However, downstream analysis was performed only on the samples taken at the end of each period because the washout period (adaptation phase) was 20 days (Appendix A). Approximately 250 mL of rumen fluid was collected in each sampling, snap-frozen into liquid nitrogen, and stored at −80 °C for later processing.

### 2.3. Feed Chemical Analysis

Sampling of feeds over the experimental period and how orts were handled were described by Thompson [23]. Composited and dried feeds were analyzed for DM (dry matter; method 934.01) as described by AOAC [24]. Both neutral (NDF; with α-amylase and sodium sulfite) and acid (ADF) detergent fiber [25] were quantified using Ankom Fiber Analyzer (Ankom Technology Corporation, Macedon, NY, USA). Starch was determined by the α-amylase method as described by Hall [26]. A Leco combustion nitrogen (N) analyzer (FP-428N Determinator, Leco Corporation, St. Joseph, MI, USA) was used to measure N content. Crude protein (CP) was calculated as N × 6.25 [19].

### 2.4. DNA Extraction

Total DNA was extracted from the ruminal content using the bead-beating method, as described by Liet al. [27]. In summary, frozen rumen content was thawed on ice, and 1 g sample was added in 15 mL falcon tubes, washed in 4 mL of TN150 buffer (10 mM Tris-HCl [pH 8.0], 150 mM NaCl), and centrifuged at 14,600× *g* for 5 min at 4 °C. Thereafter, samples were physically disrupted in a BioSpec Mini-BeadBeater-8 (BioSpec, Bartlesville, OK, USA) at 246× *g* for 3 min and subjected to a phenol/chloroform/isoamyl alcohol (25:24:1) extraction. The DNA was precipitated with cold ethanol and dissolved in nuclease-free water (30 μL). Lastly, the concentration and quality of DNA were measured using Nanodrop^®^ ND-1000 spectrophotometer (Thermo Scientific, Waltham, MA, USA). Quantitative real-time PCR (qPCR) was performed only on DNA samples exhibiting a ratio of absorbance at 260 nm to 280 nm higher than 1.8 and a ratio of 260 nm wavelength absorbance to 230 nm between 2.0 and 2.2.

### 2.5. Quantitative Real-Time PCR Analysis

The qPCR analysis was performed to estimate microbial abundances by measuring the copy numbers of targeted genes using SYBR Green chemistry (Fast SYBR^®^ Green Master Mix; Applied Biosystems, Waltham, MA, USA) on a StepOnePlusTM Real-Time PCR System (Applied Biosystems). The partial bacterial and archaeal 16S rRNA genes were amplified using U2F/U2R (5′-ACTCCTACGGGAGGCAG-3′; 5′-GACTACCAGGGTATCTAATCC-3′) [28] and uniMet1-F/uniMet1-R primer pairs (5′-CCGGAGATGGAACCTGAGAC-3′; 5′-CGGTCTTGCCCAGCTCTTATTC-3′) [29]. Protozoa and fungi were amplified using P-SSU-316F/P-SSU-539R (5′-GCTTTCGWTGGTAGTGTATT-3′; 5′-CTTGCCCTCYAATCGT WCT-3′) [30] and Fungi-F1/Fungi-R1 (5′-GAGGAAGTAAAAGTCGTAACAAGGTTTC-3′; 5′-CAAATTCACAAAGGGTAGGATGATT-3′) [31] to target 18S rRNA genes.

The qPCR experiments were performed using the following program: 95 °C for 10 min, followed by 40 cycles of 95 °C for 20 s and 62 °C for 1 min for bacteria, and 95 °C for 20 s, followed by 40 cycles of 95 °C for 3 s and 60 °C for 30s for archaea, protozoa, and fungi. A standard curve was constructed using serial dilutions of plasmid DNA containing the 16S rRNA gene sequence for bacteria and methanogens and 18S rRNA gene for protozoa and fungi. Copy numbers for each standard curve were calculated based on the following equation: (NL × A × 10^−9^)/(660 × n), in which NL was the Avogadro constant (6.02 × 10^23^), A was the molecular weight of DNA molecules (ηg), and n was the length of amplicon (bp) [32]. The copy number of 16S rRNA genes for total bacteria, total methanogens, and 18S rRNA gene for protozoa and fungi per sample was calculated using the equation of Li, Penner, Hernandez-Sanabria, Oba and Guan [27]: (QM × C × DV)/(S × W), where QM was the quantitative mean of the copy number, C was the DNA concentration of each sample (ηg/μL), DV was the dilution volume of extracted DNA (μL), S was the DNA amount subjected to analysis (ng), and W was the sample weight subjected to DNA extraction (g). The microbial abundance values were normalized by incorporating sample weight and DNA concentration into the equation above.

### 2.6. Statistical Analysis

The relationship between rumen microbial abundance (bacteria, protozoa, fungi, and archaea) and phenotypic traits (DMI, NDF intake, starch intake, CH_4_/kg DMI, FCR, and ADG) was explored through unsupervised preliminary analysis with Principal Components (PCA) using the package *mixOmics* [33]. Then, we implemented the algorithm *PC-Corr*—Principal Component-Correlation [34]—to generate discriminative functional networks for class labels (forage vs. grain) using *p*-values of Mann–Whitney tests as evaluators. Features were normalized through *z-score* (centered to have mean 0 and scaled to have standard deviation 1) or *log* (logarithm base 10 plus 1 applied to each data element to avoid problems with 0 values) before building *PC-Corr* networks in Cytoscape 3.6.0 [35].

Thereafter, microbial copy numbers (y) were analyzed according to the framework of generalized linear mixed effects models. Therefore, by considering the fixed effects of period and random effects of pens ($p_{j}$), the following framework was adopted: (1) the negative binomial distribution (NB) for microbial copy numbers, i.e., $y\;\sim \;NB\left(\lambda_{ij},\lambda_{ij}\left(1+\phi \lambda_{ij}\right)\right)$ with mean, $\lambda_{ij}$, and variance is given by the product $\lambda_{ij}\left(1+\phi \lambda_{ij}\right)$, in which ϕ is a scale parameter and (2) the log link function ($\eta_{ij}=\log\left(\lambda_{ij}\right)$) for $\eta_{ij}=\eta +\beta_{i}\tau +p_{j}$ as the linear predictor and an assumed normal (N) distribution for random effect of pen ($p_{j}$), that is, $p_{j}\;\sim \;N\left(0,\sigma^{2}\right)$. In the link function, η is the intercept, and $\beta_{i}$ is the period (τ, days) slope for a starch-rich diet (i=1) or for a predominant fibrous diet (i=2). Continuous variables such as dry matter intake, average daily gain, and residual feed intake (RFI) were analyzed as normally distributed variables, whereas feed conversion ratio (FCR) was assumed to follow the Gamma distribution. The model was fitted with the GLIMMIX procedure of SAS (SAS^®^ Studio, SAS University Edition, SAS Institute Inc., Cary, NC, USA).

## 3. Results

### 3.1. Effect of Diet on Phenotypic Traits and Rumen Microbial Counts

Diet composition significantly influenced both phenotypic traits and rumen microbial abundances (Table 1). Bulls fed grain-based diets exhibited higher dry matter intake (12.8 ± 0.11 kg/day) compared to those fed forage-based diets (10.6 ± 0.11 kg/day; *p* = 0.005).

As expected, forage NDF intake was higher in forage-fed bulls (4.4 ± 0.06 vs. 3.5 ± 0.06 kg/day; *p* < 0.001), while starch intake was higher in grain-fed bulls (4.0 ± 0.06 vs. 2.3 ± 0.06 kg/day; *p* < 0.001). The feed conversion ratio was lower (more efficient) in grain-fed bulls (5.8 ± 0.12 vs. 6.3 ± 0.13 kg feed/kg gain; *p* < 0.001), which was reflected in their higher average daily gain (1.8 ± 0.04 vs. 1.5 ± 0.04 kg/day; *p* < 0.001). The CH_4_/DMI ratio was lower in grain-fed bulls than in those fed forage-based diets (14.8 ± 0.45 vs. 18.0 ± 0.45 L CH_4_/kg DMI; *p* < 0.001).

Regarding the rumen microbial populations, bacterial abundance was lower in grain-fed bulls compared to those fed forage-based diets (1.1 ± 0.7 × 10^11^ vs. 2.8 ± 1.82 × 10^11^ copies of 16S rRNA genes/mL; *p* < 0.001). Similarly, fungal abundance was reduced in grain-fed bulls (1.5 ± 9.9 × 10^3^ vs. 3.5 ± 2.22 × 10^4^ of 18S rRNA genes copies/mL; *p* = 0.026). However, protozoan (5.3 ± 1.85 × 10^7^ vs. 4.5 ± 1.57 × 10^7^ copies of 18S rRNA genes/mL; *p* = 0.668) and methanogen (3.5 ± 8.06 × 10^8^ vs. 4.2 ± 9.7 × 10^8^ copies of 16S rRNA genes/mL; *p* = 0.247) abundance did not differ between dietary treatments.

### 3.2. Associations Between Phenotypic Traits and Rumen Microbial Counts Across Feeding Periods

Feed intake showed significant associations with bacterial abundance in the rumen (Table 2). The dry matter intake model revealed a negative relationship with bacterial copy numbers (*p* = 0.005), with a baseline bacterial population (intercept) of 28.4 ± 0.96 (log-transformed bacterial copy numbers). This relationship was characterized by negative associations during both Period 1 (−0.2 ± 0.07) and Period 2 (−0.1 ± 0.05). Similarly, forage NDF intake demonstrated an effect on bacterial abundance (*p* = 0.002, intercept = 25.3 ± 0.84), with a positive association observed in Period 1 (0.6 ± 0.18) but a weaker relationship in Period 2 (0.1 ± 0.14).

Starch intake also influenced bacterial abundance (*p* = 0.001, intercept = 27.4 ± 0.72), with consistent negative associations across both periods (Period 1: −0.5 ± 0.14; Period 2: −0.4 ± 0.11). In contrast, feed efficiency metrics showed no relationship with bacterial abundance. The feed conversion ratio (FCR) model showed no significant overall effect (*p* = 0.508), with an intercept of 11.1 ± 1.18 and no period-specific associations with bacterial abundance (Period 1: −0.1 ± 0.18; Period 2: −0.1 ± 0.17). Similarly, residual feed intake (RFI) demonstrated no relationship with bacterial abundance (*p* = 0.160). Also, the CH_4_/DMI ratio (*p* = 0.084) and average daily gain (*p* = 0.310) models showed no associations with bacterial abundance, although the CH_4_/DMI ratio model suggested a trend (*p* = 0.084) in both periods compared to the intercept.

Dry matter intake demonstrated a relationship with fungal abundance (*p* = 0.037), with a baseline population (intercept) of 12.7 ± 1.25 (log-transformed fungal copy numbers) (Table 3). This relationship was characterized by negative associations in both Period 1 (−0.2 ± 0.10) and Period 2 (−0.1 ± 0.08). In contrast, forage NDF intake showed no significant effect on fungal abundance (*p* = 0.146, intercept = 9.7 ± 1.03), despite displaying a positive association in Period 1 (0.5 ± 0.29) and a weaker relationship in Period 2 (0.1 ± 0.22) (Table 3).

Starch intake influenced fungal copy numbers (*p* = 0.027, intercept = 11.4 ± 0.81), with consistent negative associations across both periods (Period 1: −0.5 ± 0.20; Period 2: −0.3 ± 0.16) (Table 3). Feed efficiency metrics showed no significant relationships with fungal abundances. The feed conversion ratio model revealed no significant effect (*p* = 0.723, intercept = 11.5 ± 1.39), with no period-specific associations (Period 1: 0.1 ± 0.38; Period 2: 0.0 ± 0.00). Similarly, residual feed intake showed no significant relationship with fungal abundances (*p* = 0.365). The CH_4_/DMI ratio (*p* = 0.706) and average daily gain (*p* = 0.398) models demonstrated no significant associations with fungal copy numbers and no relationship across both feeding periods (Period 1: 0.0 ± 0.03; Period 2: 0.0 ± 0.04). The ADG model showed no effect on fungal abundances (*p* = 0.398), exhibiting weak negative associations in both periods (Period 1: −0.5 ± 0.55; Period 2: −0.3 ± 0.47).

None of the phenotypic traits showed associations with either protozoan or methanogen population abundances (log-transformed) across the experimental periods (Table 4 and Table 5).

### 3.3. PC-Corr Analysis of Microbial Counts and Phenotypic Traits

The PC-Corr algorithm revealed distinct clustering patterns between forage and grain-fed treatments under different normalization methods (Figure 1).

Using log normalization for the microbial copy number of targeted genes (Figure 1A,B), PC6 showed significant discrimination between dietary groups (*p* < 0.010) with high predictive performance (AUC = 0.882, AUPR = 0.871), while PC1 showed no significant separation (*p* = 0.681, AUC = 0.522, AUPR = 0.48). Under z-score normalization for the phenotypic data (Figure 1C), PC1 demonstrated group separation between diets (*p* < 0.010, AUC = 0.982, AUPR = 0.966), with PC6 also showing significant but lower discriminatory power (*p* < 0.010, AUC = 0.757, AUPR = 0.705).

Network analysis revealed distinct correlation patterns among microbial groups and phenotypic traits (Figure 1B,D). The microbial network (Figure 1B) showed positive correlations among all microbial groups, with the strongest correlation observed between fungi and protozoa (0.467), followed by bacteria and protozoa (0.301) and methanogens and protozoa (0.284), respectively. Bacteria and methanogens also showed a positive correlation (0.487).

The phenotypic trait network (Figure 1D) demonstrated clear dietary effects. Feed efficiency metrics (FCR) were negatively correlated with ADG (−0.597), and forage NDF intake (fNDF) showed a negative correlation with starch intake (−0.619). The CH_4_/DMI ratio exhibited a negative correlation with both DMI (−0.627) and starch intake (−0.577). Among grain-associated diets (shown in red), positive correlations were observed between DMI and starch intake (0.862), DMI and ADG (0.744), and starch intake and ADG (0.702).

## 4. Discussion

This study highlights the impact of feed composition on rumen microbial populations and host phenotypes, underscoring the complex interplay between diet, microbiome dynamics, and phenotypic traits in beef cattle. The application of network analysis and comparative analysis across feeding periods provided novel insights into microbial interactions and adaptation to dietary changes. Grain-based diets significantly reduced bacterial and fungal populations’ copy numbers while maintaining stable protozoan and methanogen copy numbers across feeding periods. Methane emissions per unit of dry matter intake were significantly lower in grain-fed bulls, emphasizing how feed composition influences hydrogen availability, the principal substrate for rumen methanogenesis [36].

The decline in bacterial and fungal copy numbers associated with grain-based diets reflects shifts in the rumen microbial ecology, driven by starch availability and increased feed passage rates. These results align with prior research suggesting that starch-rich diets encourage the competitive exclusion of fiber-digesting microbes [37]. However, this observed decline contrasts with studies that reported increased microbial growth under carbohydrate-rich dietary conditions [8]. This discrepancy highlights the complexity of rumen microbial adaptation, wherein factors such as feed particle size, passage rates, and microbial cross-feeding mediate microbial responses to dietary composition [37,38]. Bacterial abundance showed significant associations with dry matter (DM) and neutral detergent fiber (NDF) intakes (Table 2). These findings emphasize the pivotal role of bacteria in fiber degradation, consistent with previous studies [39]. Although fungi displayed associations with DM intake and no effect on NDF intake (Table 3), they facilitate bacterial colonization by physically disrupting plant cell walls [40,41,42]. This interdependent relationship underscores the significance of fungal–bacterial interactions in optimizing fiber utilization within the rumen.

The stability of protozoan and methanogen population across dietary treatments and phenotypic traits underscores their functional resilience within the rumen ecosystem. Inter-species associations between protozoans and methanogens are critical for methane production, as protozoa shield methanogens from being washed out of the rumen due to their slower passage rate compared to bacteria and fungi [1]. These methanogen–protozoan interactions ensure consistent methane production, even amidst fluctuations in bacterial and fungal populations. While these interactions are well-documented [43], evidence suggests that methane emissions are influenced more by the activity and composition of specific methanogen species as opposed to population size in the rumen (Zhou et al., 2011) [44]. The absence of significant correlations between protozoan or methanogen populations and phenotypic traits, such as feed conversion ratio (FCR) or average daily gain (ADG), implies that their roles in rumen metabolic processes may be indirect, acting through mechanisms like hydrogen transfer or modulation of volatile fatty acid profiles [4]. Protozoa also control bacterial numbers via predation [45], further contributing to microbial turnover and functional stability, which ultimately influence host phenotypes in response to feed composition and intake.

Methane emissions per unit DMI were considerably lower in grain-fed bulls, underscoring the role of microbial interactions in regulating hydrogen flux—a vital precursor for methanogenesis. The reduction in bacterial and fungal populations associated with grain diets likely limited hydrogen availability, thus restricting substrate supply for methanogens. This was also likely due to less hydrogen production with an increase in propionate in high-grain diets [9]. Network analysis confirmed significant correlations between bacteria and methanogens (r = 0.4873), emphasizing the importance of hydrogen producers in influencing methane output. Grain-based diets achieved a reduction in methane production per kg DMI by approximately 18% (Table 1), thereby showcasing the potential of dietary manipulation as a strategy for methane mitigation. Dietary supplementation with feed additives (e.g., 3-nitrooxypropanol) [46], combined with strategies that promote alternative hydrogen sinks, e.g., propionate synthesis [10], offers a complementary approach to dietary manipulation for reducing methane emissions from ruminants.

Comparative analysis across feeding periods revealed adaptive shifts in microbial populations during dietary transitions. Microbial communities in the rumen exhibit remarkable adaptability to changes in dietary composition, with bacteria and fungi playing distinct roles in this process [47]. The observed reduction in bacterial abundance associated with NDF intake across feeding periods (Table 2) highlights how bacterial populations adjust their metabolic strategies to accommodate shifts in substrate availability. This adaptation may involve the expansion in numbers of previously underutilized microbial species or may increase activity in functional pathways, enabling the community to maintain fiber degradation despite changing environmental conditions. No effect of fungal responses to dietary fiber suggests a secondary (as compared to bacteria) but supportive role in microbial succession (Table 2 and Table 3), underscoring the importance of microbial networks in maintaining rumen functionality. These findings align with previous studies demonstrating the need for microbial adaptation during dietary changes [39]. While significant shifts in bacterial and fungal populations were observed, no direct correlations were found between rumen microbial abundance and feed efficiency metrics such as FCR and RFI. This aligns with the concept of functional redundancy, where different microbial taxa fulfill overlapping roles to ensure metabolic stability [48]. Grain-fed bulls demonstrated improved FCR, potentially linked to enhanced nutrient digestibility associated with starch-rich diets. However, the absence of direct microbial correlations suggests that host factors, such as nutrient absorption efficiency and metabolic adaptability, may play a more dominant role in driving feed efficiency [49]. These findings underscore the importance of considering both microbial network dynamics and host physiological adaptations in optimizing feed efficiency in ruminants. Further investigations are needed to move beyond abundance data and explore the metabolic roles and functional interactions within the rumen microbial community.

To fully understand these complex interactions and translate them into practical applications requires addressing certain limitations in the current study. First, the reliance on qPCR-based microbial quantification provides a snapshot of microbial abundance but does not capture functional activity or metabolic contributions. Additionally, inter-individual variation in baseline microbiota composition may significantly influence microbial responses to dietary changes, potentially complicating the interpretation of phenotypic trait outcomes. Investigating these variations could shed light on the role of host-microbiome individuality in driving phenotypic traits. Finally, validating these findings across diverse production systems and cattle breeds will be essential to develop robust strategies for enhancing ruminant productivity and sustainability.

## 5. Conclusions

This study highlights the impact of feed composition on rumen microbial dynamics and methane emissions. While protozoan and methanogen populations remained stable across dietary treatments, bacterial and fungal copy numbers were significantly reduced in grain-fed bulls, reflecting a microbial ecological shift driven by starch availability. As expected, methane emissions per unit of dry matter intake were significantly lower in grain-fed bulls, suggesting that the reduced emissions may be attributed to the movement of hydrogen into alternative sinks, typically linked to bacterial and fungal activity, rather than direct alterations in protozoan or methanogen population densities.

No direct correlations were found between microbial abundance and feed efficiency metrics, suggesting that metabolic function and microbial network dynamics, in addition to taxonomic composition, play a more significant role in driving feed efficiency. Inter-period feeding shifts revealed adaptive microbial responses to dietary transitions between forage- and grain-based diets, particularly in bacterial and fungal populations. These findings demonstrate the adaptability of the rumen microbiome to feeding conditions and underscore its potential as a target for strategies aimed at improving ruminant productivity and sustainability.

## Figures and Tables

**Figure 1 microorganisms-13-00310-f001:**
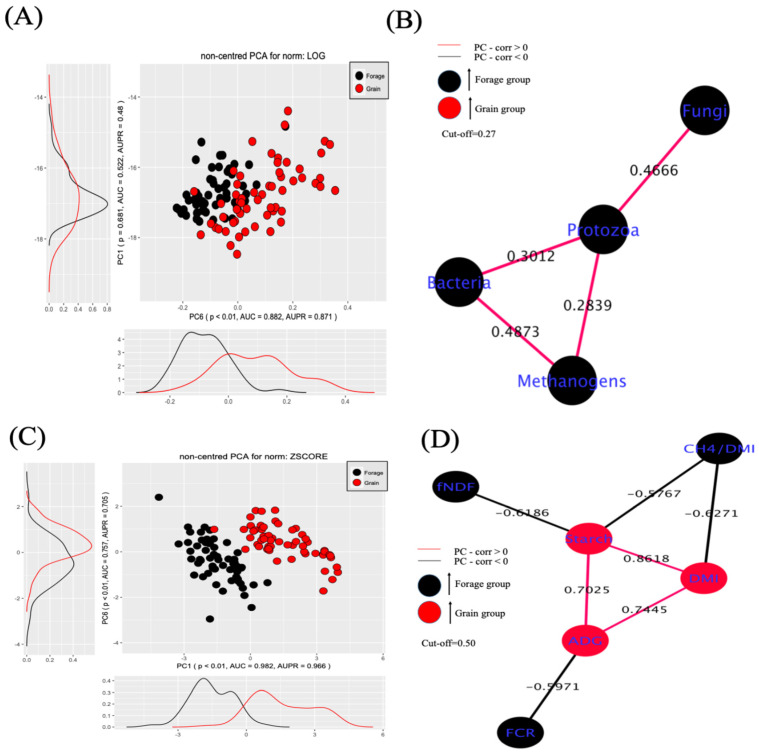
Rumen microbial population and phenotypic traits in response to diet changes. Linear dimensionality reduction by PC-Corr separated rumen microbes (**A**) and phenotypic traits (**C**) recorded in cattle fed forage (black) or grain diets (red). PC-Corr detected differences (Mann–Whitney test, *p* < 0.01) in rumen microbes between forage and grain diets along principal component 6 (PC6), whose loadings were used to build a microbial network at a cut-off (Pearson correlation) of 0.27 (**B**). Differences (Mann–Whitney test, *p* < 0.01) in phenotypic traits were also observed for forage and grain diets along PC1 and PC6, whose loadings were used for the construction of a network at a cut-off of 0.50 (**D**).

**Table 1 microorganisms-13-00310-t001:** Predictors, mean ± standard errors (SE), and 95% confidence intervals (Upper–Lower Limits) for the phenotype data and microbial counts in the rumen of bulls fed either forage- or grain-based diets.

	Diets (Mean ± SE)		
Predictors	Forage (Upper–Lower Limits)	Grain (Upper–Lower Limits)	*p*-Value
Phenotype data **(Unit)**			
Dry matter intake, (kg/day)	10.6 ± 0.11 (10.1–11.1)	12.8 ± 0.11 (12.3–13.3)	0.005
Forage NDF intake, (kg/day)	4.4 ± 0.06 (4.2–4.5)	3.5 ± 0.06 (3.3–3.6)	<0.001
Starch intake, (kg/day)	2.3 ± 0.06 (2.2–2.4)	4.0 ± 0.06 (3.9–4.1)	<0.001
Feed conversion ratio, (kg feed/kg gain)	6.3 ± 0.13 (6.0–6.7)	5.8 ± 0.12 (5.5–6.1)	<0.001
CH_4_/DMI ratio, (Liters CH_4_/kg DMI)	18.0 ± 0.45 (17.1–18.9)	14.8 ± 0.45 (13.9–15.7)	<0.001
Average daily gain, (kg/day)	1.5 ± 0.04 (1.3–1.6)	1.8 ± 0.04 (1.7–2.0)	<0.001
Microbial copy numbers			
Bacteria ^1^	2.8 ± 1.82 × 10^11^ (3.9 × 10^10^–2.0 × 10^12^)	1.1 ± 0.7 × 10^11^ (1.6 × 10^10^–8.3 × 10^11^)	<0.001
Fungi ^2^	3.5 ± 2.22 × 10^4^ (5.4 × 10^3^–2.3 × 10^5^)	1.5 ± 9.9 × 10^3^ (2.4 × 10^3^–1.0 × 10^5^)	0.026
Protozoa ^3^	4.5 ± 1.57 × 10^7^ (1.7 × 10^5^–1.1 × 10^8^)	5.3 ± 1.85 × 10^7^ (1.7 × 10^7^–1.1 × 10^8^)	0.668
Methanogens ^4^	4.2 ± 9.7 × 10^8^ (2.1 × 10^8^–8.3 × 10^8^)	3.5 ± 8.06 × 10^8^ (1.8 × 10^8^–6.9 × 10^8^)	0.247

^1,4^ Copy number of 16S rRNA (Mean ± SE)/mL of rumen fluid. ^2,3^ Copy number of 18S (Mean ± SE)/mL of rumen fluid.

**Table 2 microorganisms-13-00310-t002:** Predictors, estimates (log-transformed), and confidence intervals derived from the selected model for bacterial copy numbers in the rumen of bulls fed forage- or grain-based diets ^1^.

Predictors	Estimates	95% Confidence Intervals (Upper–Lower Limits)	*p*-Value
DMI **—Dry matter intake**			
Intercept	28.4 ± 0.96	26.4–30.5	0.005
DMI ∗ Period 1	−0.2 ± 0.07	−0.3–[−0.09]	
DMI ∗ Period 2	−0.1 ± 0.05	−0.3–[−0.08]	
fNDF **—Forage NDF intake**			
Intercept	25.3 ± 0.84	23.3–27.2	0.002
fNDF ∗ Period 1	0.6 ± 0.18	0.2–1.02	
fNDF ∗ Period 2	0.1 ± 0.14	−0.1–0.44	
Starch **intake**			
Intercept	27.4 ± 0.72	25.6–29.2	0.001
Starch ∗ Period 1	−0.5 ± 0.14	−0.7–[−0.24]	
Starch ∗ Period 2	−0.4 ± 0.11	−0.6–[−0.19]	
FCR **—Feed conversion ratio**			
Intercept	11.1 ± 1.18	8.7–13.5	0.508
FCR ∗ Period 1	−0.1 ± 0.18	−0.5–0.17	
FCR ∗ Period 2	−0.1 ± 0.17	−0.4–0.19	
RFI **—Residual feed intake**			
Intercept	25.9 ± 0.56	24.1–27.7	0.160
RFI ∗ Period 1	−0.2 ± 0.23	−0.7–0.22	
RFI ∗ Period 2	−0.2 ± 0.14	−0.5–0.05	
**CH4/DMI ratio**			
Intercept	24.9 ± 0.74	23.2–26.6	0.084
CH4/DMI ∗ Period 1	0.0 ± 0.02	0.0–0.10	
CH4/DMI ∗ Period 2	0.0 ± 0.03	0.0–0.13	
**ADG—Average daily gain**			
Intercept	25.9 ± 0.99	23.8–27.9	0.310
ADG ∗ Period 1	−0.7 ± 0.75	−2.2–0.72	
ADG ∗ Period 2	0.0 ± 0.00	0.0–0.0	

^1^ Rumen fluid samples were analyzed using qPCR on Days 80 (Period 1) and 180 (Period 2).

**Table 3 microorganisms-13-00310-t003:** Predictors, estimates (log-transformed), and confidence intervals derived from the selected model for fungal copy numbers in the rumen of bulls fed forage- or grain-based diets ^1^.

Predictors	Estimates	95% Confidence Intervals (Upper–Lower Limits)	*p*-Value
DMI **—Dry matter intake**			
Intercept	12.7 ± 1.25	10.2–15.25	0.037
DMI∗Period 1	−0.2 ± 0.10	−0.4–[−0.05]	
DMI∗Period 2	−0.1 ± 0.08	−0.3–[−0.01]	
fNDF **—Forage NDF intake**			
Intercept	9.7 ± 1.03	7.5–11.8	0.146
fNDF∗Period 1	0.5 ± 0.29	0.0–1.16	
fNDF∗Period 2	0.1 ± 0.22	−0.3–0.58	
Starch **intake**			
Intercept	11.4 ± 0.81	9.6–13.2	0.027
Starch∗Period 1	−0.5 ± 0.20	−0.9–[−0.12]	
Starch∗Period 2	−0.3 ± 0.16	−0.6–[−0.01]	
FCR **—Feed conversion ratio**			
Intercept	11.5 ± 1.39	8.7–14.3	0.723
FCR∗Period 1	0.1 ± 0.38	−0.6–0.88	
FCR∗Period 2	0.0 ± 0.00	0.0–0.00	
RFI **—Residual feed intake**			
Intercept	10.0 ± 0.50	8.4–11.7	0.365
RFI∗Period 1	−0.3 ± 0.36	−1.0–0.36	
RFI∗Period 2	−0.2 ± 0.22	−0.6–0.21	
**CH4/DMI ratio**			
Intercept	10.4 ± 0.86	8.6–12.2	0.706
CH4/DMI∗Period 1	0.0 ± 0.03	−0.1–0.05	
CH4/DMI∗Period 2	0.0 ± 0.04	−0.1–0.07	
**ADG—Average daily gain**			
Intercept	10.8 ± 1.00	8.7–12.8	0.398
ADG∗Period 1	−0.5 ± 0.55	−1.6–0.54	
ADG∗Period 2	−0.3 ± 0.47	−1.2–0.61	

^1^ Rumen fluid samples were analyzed using qPCR on Days 80 (Period 1) and 180 (Period 2).

**Table 4 microorganisms-13-00310-t004:** Predictors, estimates (log-transformed), and confidence intervals derived from the selected model for protozoan copy numbers in the rumen of bulls fed forage- or grain-based diets ^1^.

Predictors	Estimates	95% Confidence Intervals (Lower–Upper Limits)	*p*-Value
DMI **—Dry matter intake**			
Intercept	8.9 ± 1.25	6.4–11.46	0.365
DMI∗Period 1	0.0 ± 0.11	−0.1–0.29	
DMI∗Period 2	0.0 ± 0.09	−0.1–0.21	
fNDF **—Forage NDF intake**			
Intercept	18.6 ± 0.94	16.7–20.62	0.422
fNDF∗Period 1	0.0 ± 0.31	−0.5–0.67	
fNDF∗Period 2	−0.3 ± 0.24	−0.8–0.17	
Starch **intake**			
Intercept	9.1 ± 0.69	7.7–10.60	0.401
Starch∗Period 1	0.1 ± 0.22	−0.2–0.62	
Starch∗Period 2	0.0 ± 0.18	−0.3–0.41	
FCR **—Feed conversion ratio**			
Intercept	10.7 ± 1.06	8.6–12.88	0.215
FCR∗Period 1	−0.1 ± 0.22	−0.5–0.16	
FCR∗Period 2	−0.2 ± 0.16	−0.5–0.10	
RFI **—Residual feed intake**			
Intercept	9.5 ± 0.27	8.6–10.39	0.417
RFI∗Period 1	−0.4 ± 0.35	−1.1–0.23	
RFI∗Period 2	−0.0 ± 0.21	−0.4–0.42	
**CH4/DMI ratio**			
Intercept	9.8 ± 0.75	8.3–11.39	0.235
CH4/DMI∗Period 1	0.0 ± 0.03	0.0–0.06	
CH4/DMI∗Period 2	0.0 ± 0.04	−0.1–0.05	
**ADG—Average daily gain**			
Intercept	8.5 ± 0.89	6.7–10.28	0.203
ADG∗Period 1	0.7 ± 0.54	−0.3–1.83	
ADG∗Period 2	0.4 ± 0.46	−0.4–1.37	

^1^ Rumen fluid samples were analyzed using qPCR on Days 80 (Period 1) and 180 (Period 2).

**Table 5 microorganisms-13-00310-t005:** Predictors, estimates (log-transformed), and confidence intervals derived from the selected model for methanogen copy numbers in the rumen of bulls fed forage- or grain-based diets ^1^.

Predictors	Estimates	95% Confidence Intervals (Lower–Upper Limits)	*p*-Value
DMI **—Dry matter intake**			
Intercept	20.1 ± 0.54	19.0–21.26	0.530
DMI∗Period 1	0.0 ± 0.04	−0.1–0.05	
DMI∗Period 2	0.0 ± 0.04	−0.1–0.05	
fNDF **—Forage NDF intake**			
Intercept	19.8 ± 0.43	18.9–20.73	0.170
fNDF∗Period 1	0.2 ± 0.12	0.0–0.48	
fNDF∗Period 2	0.0 ± 0.09	−0.1–0.18	
Starch **intake (?)**			
Intercept	20.0 ± 0.33	19.3–20.77	0.324
Starch∗Period 1	−0.1 ± 0.09	−0.3–0.06	
Starch∗Period 2	0.0 ± 0.07	−0.2–0.08	
FCR **—Feed conversion ratio**			
Intercept	19.1 ± 0.48	18.2–20.16	0.313
FCR∗Period 1	0.0 ± 0.07	0.0–0.24	
FCR∗Period 2	0.1 ± 0.07	0.0–0.24	
RFI **—Residual feed intake**			
Intercept	19.7 ± 0.17	19.2–20.35	0.672
RFI∗Period 1	0.1 ± 0.15	−0.1–0.44	
RFI∗Period 2	0.0 ± 0.09	−0.2–0.16	
**CH4/DMI ratio**			
Intercept	19.5 ± 0.34	18.8–20.29	0.550
CH4/DMI∗Period 1	0.0 ± 0.01	0.0–0.04	
CH4/DMI∗Period 2	0.0 ± 0.02	0.0–0.05	
**ADG—Average daily gain**			
Intercept	20.3 ± 0.41	19.5–21.21	0.211
ADG∗Period 1	−0.4 ± 0.24	−0.8–0.06	
ADG∗Period 2	−0.3 ± 0.20	−0.7–0.10	

^1^ Rumen fluid samples were analyzed using qPCR on days 80 (Period 1) and 180 (Period 2).

## Data Availability

The original contributions presented in this study are included in the article/Appendix A. Further inquiries can be directed to the corresponding author.

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
