# Peer review of "Impact of Feed Composition on Rumen Microbial Dynamics and Phenotypic Traits in Beef Cattle"

_microorganisms, 2025, doi:10.3390/microorganisms13020310_

Round 1
Reviewer 1 Report
Comments and Suggestions for Authors
Line 90, I don't believe this is a crossover design because the cattle in the four groups are not from the same batch; instead, each group received different feeding methods (Forage and Forage, Forage and Grain, Grain and Forage, Grain and Grain over two periods). This should be considered a randomized block design. It is recommended to have a professional statistical analysis expert carefully review and analyze it.
Line 98-99 Could you bring me more details about Residual feed intake (RFI) estimation?
Line 101 When is methane measured during the experiment, on which day? Is there an adaptation period for methane measurement? With only one day dedicated to methane measurement per period and no repeated days, the error in methane production may be significant.
Line 160 Statistical analysis
I don't believe this is a crossover design, so please have a dedicated statistical expert verify it and double-check for accuracy.
Table S2 What is the basis for formulating the recipe? Does it meet the nutritional needs of Angus bulls? The cumulative percentage of the diet is not 100%. Please carefully check it.
Author Response
Comment 1: [Line 90, I don't believe this is a crossover design because the cattle in the four groups are not from the same batch; instead, each group received different feeding methods (Forage and Forage, Forage and Grain, Grain and Forage, Grain and Grain over two periods). This should be considered a randomized block design. It is recommended to have a professional statistical analysis expert carefully review and analyze it.]
Reply 1: [Thank you for your observation regarding the experimental design in our study! We completely agree with your assessment that this is indeed a randomized block design. During manuscript preparation, the term "crossover design" was mistakenly carried over from another manuscript template. As you can observe, our experimental design was analyzed appropriately using the effect of pen (block), as described in the statistical analysis section.
We have corrected this mislabeling in the revised manuscript. This correction aligns with the statistical analysis and the actual setup of the experiment, where cattle were housed in separate pens and assigned distinct feeding methods across the two periods.
We would also like to assure the reviewer that the statistical analysis remains entirely accurate. The data were analyzed using a generalized linear mixed model, accounting for the random effect of pen (block) and fixed effects as originally detailed in the manuscript. Since the statistical analysis is accurate and correct, the initial terminology oversight does not affect the validity of our findings.]
Comment 2: [Line 98-99 Could you bring me more details about Residual feed intake (RFI) estimation?]
Comment 3: [Line 101 When is methane measured during the experiment, on which day? Is there an adaptation period for methane measurement? With only one day dedicated to methane measurement per period and no repeated days, the error in methane production may be significant.]
Replies 2 and 3: [Residual feed intake (RFI) was estimated as the difference between the observed feed intake and the predicted feed intake, which was derived using a linear regression model based on metabolic body weight (BW⁰·⁷⁵) and average daily gain (ADG). This approach aligns with established methodologies for evaluating feed efficiency in cattle. The RFI measurement period spanned 79 days (with 76 days of valid intake data). Days were excluded if feed disappearance was less than 95%, ensuring the accuracy and validity of intake data.
To ensure consistency, intake data were collected using the GrowSafe® system, which enabled precise monitoring of individual feed intake throughout the trial. Methane measurements were deliberately avoided during the RFI period to eliminate potential disruptions to normal feed intake behavior.
Reply to Comment 3:
Methane production was measured using the sulfur hexafluoride (SF₆) tracer gas technique, which allows unrestricted animal movement and closely mimics real-world production environments. Methane measurements were conducted during the adaptation and post-RFI periods, avoiding the RFI period to prevent interference with feed intake data collection.
Methane Measurement Timeline:
-
Adaptation Period:
- Methane measurements were conducted on four separate occasions during the adaptation period to allow animals to acclimate to the experimental setup: Days -12, -9, -6, and -2 relative to the start of the RFI period.
-
Post-RFI Period:
- Methane was measured again on three consecutive days following the RFI period to evaluate emissions under steady-state conditions: Days 80, 82, and 85.
Addressing Measurement Frequency and Potential Error:
While repeated measurements per animal within a period would reduce error, the frequency of methane measurements in our study was higher than many other studies using the SF₆ technique, which often rely on a single measurement per period. By conducting multiple measurements during the adaptation and post-RFI periods, we aimed to capture methane production variability while balancing the labor-intensive nature of this technique. The SF₆ method was chosen because it allows animals to move freely, reflecting real production environments more accurately than chamber-based approaches.
We have included all those details in the manuscript - thanks].
Comment 4: [Line 160 Statistical analysis
I don't believe this is a crossover design, so please have a dedicated statistical expert verify it and double-check for accuracy.]
Reply 4: [The reviewer is absolutely correct in identifying the issue with the terminology. As explained above, we have corrected the term to properly reflect that the study used a randomized block design, not a crossover design. This correction ensures consistency between the experimental design and its description in the manuscript, as well as the statistical analysis.
We would like to assure the reviewer that the statistical analysis is entirely accurate and was conducted appropriately for a randomized block design. The data were analyzed using a generalized linear mixed model, accounting for the random effect of pen (block) and other fixed effects as described in the manuscript. While the initial mislabeling was an oversight, the statistical framework remains valid.]
Comment 5: [Table S2 What is the basis for formulating the recipe? Does it meet the nutritional needs of Angus bulls? The cumulative percentage of the diet is not 100%. Please carefully check it.]
Reply 5: [Thank you for your observation regarding the diet formulation and the cumulative percentages in Table S2. The diets were formulated to meet the nutritional requirements of Angus bulls in accordance with the NRC (2016) guidelines for beef cattle.
We have carefully reviewed and corrected the ingredient percentages in Table S2 to ensure they sum to 100%, as you correctly observed].
Reviewer 2 Report
Comments and Suggestions for Authors
the present manuscript is interesting and gives a good perspective on the interaction between feed composition and the rumen microbial population, as well as some phenotypic traits.
the introduction appears well structured and sufficient bibliography is provided with a good overview on the matter.
matherials and methods are adequately descripted and reproducible.
results are interesting and broadly described and justified, as well as tables provided helpful for the understanding of the results.
discussion section is sufficient in bibliographical notes as well and the results obtained are adequately discussed and compared to precedent findings on the matter.
Author Response
Thank you for your positive and encouraging feedback on our manuscript. We are pleased to hear that you found the study interesting and appreciated the structure, content, and thoroughness of the introduction, methods, results, and discussion sections. Your thoughtful review motivates us to continue exploring these important interactions in ruminant nutrition and microbiology.
Round 2
Reviewer 1 Report
Comments and Suggestions for Authors
In lines 99-114, please add related references for the methods of Residual Feed Intake (RFI) calculation, the linear regression model, and the SF6 technique.
Author Response
Comment 1: [In lines 99-114, please add related references for the methods of Residual Feed Intake (RFI) calculation, the linear regression model, and the SF6 technique].
Reply 1: [We have revised the text accordingly and added related references for the methods of Residual Feed Intake (RFI) calculation, the linear regression model, and the SF₆ technique. The following references have been included in the revised manuscript:
-
Koch, R.M.; Swiger, L.A.; Chambers, D.; Gregory, K.E. Efficiency of feed use in beef cattle. J. Anim. Sci. 1963, 22, 486–494. https://doi.org/10.2527/jas1963.222486x.
-
Basarab, J.A.; Price, M.A.; Aalhus, J.L.; Okine, E.K.; Snelling, W.M.; Lyle, K.L. Residual feed intake and body composition in young growing cattle. Can. J. Anim. Sci. 2003, 83, 189–204. https://doi.org/10.4141/A02-065.
-
Berry, D.P.; Crowley, J.J. Residual intake and body weight gain: A new measure of efficiency in growing cattle. J. Anim. Sci. 2012, 90, 109–115. https://doi.org/10.2527/jas.2011-4245.
-
Berndt, A.; Boland, T.; Deighton, M.; Gere, J.; Grainger, C.; Hegarty, R.; Iwaasa, A.; Koolaard, J.; Lassey, K.; Luo, D. Guidelines for use of sulphur hexafluoride (SF₆) tracer technique to measure enteric methane emissions from ruminants. Ministry of Primary Industries: Wellington, New Zealand 2014, pp. 1–66.
-
Deighton, M.H.; Williams, S.R.; Hannah, M.C.; Eckard, R.J.; Boland, T.M.; Wales, W.J.; Moate, P.J. A modified SF₆ tracer technique to measure enteric methane emissions from lactating dairy cows. Anim. Feed Sci. Technol.2014, 197, 47–63. https://doi.org/10.1016/j.anifeedsci.2014.07.002.
Thank you for your valuable feedback, which has enhanced the quality of our work].